Resource

# Myeloma immunoglobulin rearrangement and translocation detection through targeted capture sequencing

Signy Chow[1,2,3] , Olena Kis[1], David T Mulder[1] , Arnavaz Danesh[1], Jeff Bruce[1], Ting Ting Wang[1,3], Donna Reece[1,3], Nizar Bhalis[4], Paola Neri[4], Peter JB Sabatini[1,3] , Jonathan Keats[5] , Suzanne Trudel[1,3,*], Trevor J Pugh[1,3,6,*]

**Multiple myeloma is a plasma cell neoplasm characterized by clonal immunoglobulin V(D)J signatures and oncogenic immunoglobulin gene translocations. Additional subclonal genomic changes are acquired with myeloma progression and therapeutic selection. PCR-based methods to detect V(D)J rearrangements can have biases introduced by highly multiplexed reactions and primers undermined by somatic hypermutation, and are not readily extended to include mutation detection. Here, we report a hybrid-capture approach (CapIG-seq) targeting the 3′ and 5′ ends of the V and J segments of all immunoglobulin loci that enable the efficient detection of V(D)J rearrangements. We also included baits for oncogenic translocations and mutation detection. We demonstrate complete concordance with matched whole-genome sequencing and/or PCR clonotyping of 24 cell lines and report the clonal sequences for 41 uncharacterized cell lines. We also demonstrate the application to patient specimens, including 29 bone marrow and 39 cell-free DNA samples. CapIG-seq shows concordance between bone marrow and cfDNA blood samples (both contemporaneous and follow-up) with regard to the somatic variant, V(D)J, and translocation detection. CapIG-seq is a novel, efficient approach to examining genomic alterations in myeloma.**

## Introduction

Recurrent translocations of immunoglobulin genes occur in ~40% of multiple myeloma (Bergsagel & Kuehl, 2015). These "primary" immunoglobulin translocations are present in similar proportions in myeloma and in the premalignant condition, monoclonal gammopathy of unknown significance and tend to remain stable through the course of disease evolution (Morgan et al, 2013; Binder et al, 2016). Primary translocations are thought to occur during errors in the normal processes of B-cell development—V(D)J recombination, class switch recombination (CSR), and somatic hypermutation (SHM)—and are thus mediated by recombination activation genes and activation-induced cytidine deaminase (Gonzalez et al, 2007). Major groups of immunoglobulin heavy chain (*IGH*) translocations include *IGH-MAF*, *IGH-FGFR3/WHSC1*, and *IGH-*cyclin D families, occurring in 10–15%, 14–20%, and 15–20% of myeloma, respectively. Most of the primary translocations occur through CSR errors; however, 21–25% of t(11;14) and t(14;20) translocations have been demonstrated to occur through errors in D-J recombination (Walker et al, 2013). FISH is commonly used to detect primary translocations and inform clinical decision-making.

Secondary events resulting in myeloma progression include the acquisition of MYC translocations, copy-number changes, and mutations, particularly relating to the dysregulation of cyclin D and cell signaling, the activation of the NF-kB pathway, and mutations in the RAS/RAF pathway (Morgan et al, 2012). Therapeutic selection pressures subsequently result in the reduction in specific myeloma subclones with an expansion of others, leading to observed patterns of clonal tiding and clonal evolution with subsequent treatments (Keats et al, 2012). V(D)J rearrangements and primary translocations provide a clonal background marker for monitoring, whereas the detection of secondary rearrangements, mutations, and epigenetic alterations (Gkotzamanidou et al, 2014) allows for monitoring of subclonality, disease evolution, and response to targeted therapy.

Significant advances in myeloma therapy in the past decade have resulted in high rates of deep responses and improvements in survival such that the minimal residual disease (MRD) is increasingly being used as a surrogate endpoint and has been incorporated into international guidelines for myeloma response assessment (Kumar et al, 2016). Current strategies for MRD detection include multiparameter flow cytometry (MFC) (Flores-Montero et al, 2017) and allele-specific or multiplex PCR of

---

[1]University Health Network, Toronto, Canada   [2]Sunnybrook Health Sciences Centre, Toronto, Canada   [3]University of Toronto, Toronto, Canada   [4]University of Calgary, Calgary, Canada   [5]Translational Genomics Research Institute, City of Hope, AZ, USA   [6]Ontario Institute for Cancer Research, Toronto, Canada

Correspondence: trevor.pugh@utoronto.ca; suzanne.trudel@uhn.ca
*Suzanne Trudel and Trevor J Pugh contributed equally to this work.

immunoglobulin loci followed by high-throughput sequencing (Puig et al, 2014).

To track the progression of multiple myeloma over time, we developed and validated a hybrid-capture sequencing assay (CapIG-seq) targeting the 3′ and 5′ ends of the V and J segments subject to V(D)J rearrangements of all immunoglobulin loci and hotspots of recurrent primary translocations. A variation of this technique, tiling the full immunoglobulin locus, has been demonstrated to reliably detect IGH gene rearrangements in tumor and cfDNA in patients with malignant B-cell lymphomas (He et al, 2011). We combined our assay with previously established targeted gene sequencing assays to enable the simultaneous detection of clonal markers, primary translocations, and oncogenic mutations from a single aliquot of DNA and applied our assay to cell lines, bone marrow cells, and cfDNA from patients with myeloma. This approach allows for rapid and accurate determination of multiple genomic alterations traditionally approached by multiple labor-intensive methods.

# Results

We designed and tested an assay and workflow that would allow for the simultaneous detection of multiple genomic alterations (Fig 1). Specific details regarding bioinformatics tools and analysis parameters are provided in Fig S1 and Table S1. This study was approved by the local institutional review board. Patients consented to the use of their biological samples.

## V(D)J detection

We designed a targeted IG capture panel using a strategy we employed for T-cell receptor sequencing (Mulder et al, 2018). The ImmunoGeneTics (IMGT) reference database annotates known variants of immunoglobulin V, D, and J genes (Lefranc et al, 1999). Probes were designed to hybridize to the 3′ ends of all IMGT-annotated V genes and the 5′ ends of J genes to maximize the likelihood of capturing DNA fragments spanning the V-J or V(D)J rearrangement junction and incorporated barcoded library preparation to improve variant calling (Fig 2 and Table S2).

To validate the IG-directed panel, we first tested 24 cell lines (23 myeloma and one B-lymphoblast) from 16 donors with available whole-genome sequencing (WGS) data. Cell lines were sequenced on the Illumina NextSeq 500 with the 150-bp paired-end application to achieve 2,500–3,000X coverage. The MiXCR (Bolotin et al, 2015) algorithm was used to call V(D)J rearrangements (Table S3). After filtering for specificity, clonal fraction of >10%, and minimum clone count of 50 (Fig S2), 48 V(D)J rearrangements with putative CDR3 sequences were found by targeted capture sequencing (Table S4). An additional 41 myeloma cell lines without available WGS data were sequenced; each cell line had at least one (median 2, range 1–6) unique V(D)J rearrangement identified (Table S4). All V(D)J rearrangements in cell lines were validated by manual review in both the targeted sequencing and WGS data (Fig S3A).

The MiXCR analysis of WGS data called 32 V(D)J rearrangements, identical to those found in targeted sequencing data. The manual inspection of the remaining 16 V(D)J rearrangements called by

targeted sequencing confirmed an additional 12 rearrangements evident at low coverage (<5 reads) in WGS (Table S4). Biological replicates between cell lines derived from the same donor yielded identical CDR3 sequence results, validating targeted capture sequencing as an accurate and reliable technique to discover clonal V(D)J rearrangements (Fig S3B and Table S4).

We also compared IGHV rearrangements and CDR3 sequences identified by our assay with those identified by a PCR-based assay, LymphoTrack, for 13 cell lines known to have IGHV-rearranged alleles by our assay. LymphoTrack identified 14 IGH rearrangements in 13 cell lines. The IGHV-J alleles were the same as those identified by CapIG-seq in 13 of 14 cases. Identical CDR3 sequences were identified in all cases, although the clonal fraction was variable between assays. There were also low-frequency (<1%) rearrangements detected unique to each assay that may have been because of artifact or sequencing error. CapIG-seq identified a rearrangement in the ALMC1 cell line not found by LymphoTrack that was filtered out by specificity. IGHD, IGKV, and IGLV genes are not identified by the LymphoTrack assay and so could not be compared. However, CapIG-seq yields identical results to those of the LymphoTrack assay in detecting IGH gene rearrangements that are validated by WGS sequencing (Tables 1, S5, and S6).

### Translocation detection and breakpoints (validation)

Probes designed to hybridize to regions containing translocation hotspots within the immunoglobulin genes were designed to detect primary rearrangements in multiple myeloma and obtained from a literature review of known breakpoint regions where possible (Walker et al, 2013; Affer et al, 2014; Bolli et al, 2016). Constant (C gene) probes were also baited, because errors within CSR within these regions are a known mechanism of illegitimate rearrangement in myeloma (Walker et al, 2013). We used two bioinformatics tools to detect and visualize translocation breakpoints, Break-Dancer, and the in-house algorithm, CluMP. Candidate rearrangements were manually inspected, and neither tool was more sensitive or specific for true rearrangements.

Of the known t(4;14) FGFR3/MMSET family translocations, 17 of 20 were called by at least one of BreakDancer or CluMP and were manually verified. Reads supporting the remaining three translocations were found upon manual verification in Integrated Genome Viewer (IGV) (Thorvaldsdottir et al, 2013), illustrating a need to tune automated calling methods for our panel. All IGH breakpoints for t(4;14) translocations were located 5′ to IGH-J regions presumably resulting from errors in CSR. Approximately 50% of breakpoints were located between IGH-J regions and the first exon of IGHM; the remainder were distributed upstream of IGHG1 or IGHA, consistent with previous reports (Walker et al, 2013) (Fig S4 and Tables S7–S9).

For cyclin D family translocations, 10 of 14 t(11;14) were found by manual inspection, two were equivocal, and the remaining two could not be found. Of the 10 translocations present or equivocal, 9 were called by BreakDancer, 9 were called by CluMP, and 8 were called by both algorithms. One of two samples with CCND2 t(12;14) translocations and two of three known CCND3 t(6;14) translocations were detectable. We also uncovered a previously unreported t(6;14) translocation cell line SKMM1, which was confirmed by manual review. Most of the CCND1 translocation partner breakpoints were found within and just

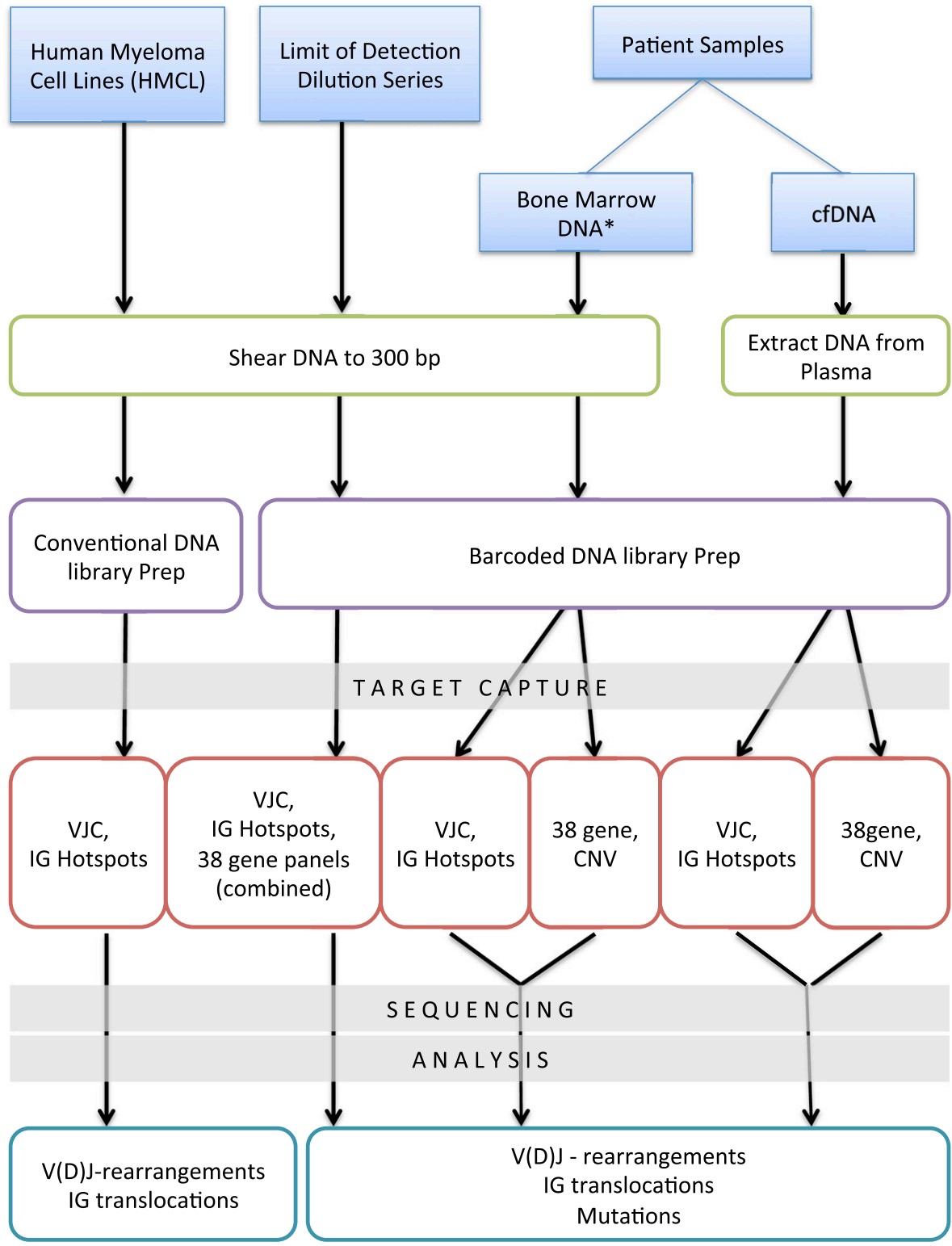

**Figure 1. Experimental Workflow.**
Development and validation of target capture sequencing platform. Genomic DNA from myeloma cell lines (left) and normal donor (middle) was sheered to 300 bp for both cell line and limit-of-dilution experiments. For myeloma cell lines, conventional DNA library prep was used to create DNA libraries. To improve sensitivity for limit-of-dilution series (middle) and patient samples (right), barcoded DNA library prep was used. Targeted capture experiments were performed by pooling targeted capture probe sets either before DNA capture or during sequencing with similar results (not shown). VJC = combination of V-gene, J-gene, and C-gene probes, IgHotspots = known translocation breakpoints within the IGH, IGK, and IGL loci. 38 gene = probes from 38 genes of interest in myeloma. CNV = copy-number probes for chromosomes

upstream of *IGHM*, but also within *IGHE* (U266) and *IGHG4* (L363), within J regions (H1112), and between J and V regions (FLAM76). *CCND3* translocations were found within *IGHG2* (KMM) and within or directly upstream of *IGHM* (SKMM1). The *IGH* translocation breakpoint for *CCND2* in AMO1 is found between *IGHM* and *IGH-J* regions (Fig S4).

*c-MAF* translocations t(14;16) were found in 10 of 11 samples, nine of which were called by BreakDancer and 7 of which were called by CluMP. *C-MAF-IGL* translocations were known from WGS of three cell lines, and all were detected by a targeted panel. Six cell lines had known *MAFB* translocations, and four were detectable by targeted sequencing. Most of the *c-MAF* translocations were found between the *IGH-J* regions and *IGHM*; 2 of 11 cell lines had translocation breakpoints within the *IGH-J* region. In contrast, two of three *MAFB* translocations (ALMC1 and ALMC2) occurred within the *IGH-J* region (Fig S4).

In summary, 20 of 20 (100%) *FGFR/MMSET* family translocations, 14 of 20 (70%) *cyclin D* family translocations, and 17 of 20 (85%) *MAF* family translocations were detectable by targeted capture sequencing (Table 2). Together, 85% of known translocations were detected by targeted sequencing and confirmed with manual verification. In six instances, manual verification was needed to verify the presence of the translocation. The sensitivity of the assay was variable depending on the translocation, and low numbers of cell lines with more uncommon translocations preclude an accurate estimation of performance in these cases. Of the common translocations in myeloma, the sensitivity ranges from 71% for t(11;14) to 100% for t(4;14) (Table 2). As a result of the nature of the assay, which includes manual verification, there are no identified false positives, and therefore, the specificity approaches 100%. These data illustrate the power of targeted *IG* sequencing to isolate clinically relevant rearrangements, and highlight an opportunity for further development of bioinformatics tools to analyze the resulting sequencing data.

### Limit of detection for translocations, V(D)J rearrangements, and somatic mutations

Having validated the individual V(D)J and translocation hotspot panels against whole-genome data, we next combined these panels with our previously developed somatic variant detection panel. We sought to determine the limit of detection with this approach using DNA from myeloma cell lines (KMS11, RPMI-8226, and MM1S) diluted into DNA from PBMCs of one healthy volunteer.

The KMS11 cell line carries both t(4;14) and t(14;16) translocations. Three distinct breakpoints were identified, two for the balanced t(4;14) translocation and one for t(14;16). With genomic DNA sheared to 300 bp, one t(4;14) breakpoint was identified at the 1/10 and 1/100 dilutions by BreakDancer; however, the manual review confirmed read pairs supporting this translocation breakpoint at the $1/10^3$ dilution and a single read at the $1/10^5$ dilution, all with identical soft-clipped reads (reads with a breakpoint where sequence deviates from reference are identical to those supporting the translocation in less dilute samples). The second breakpoint for t(4;14) was called at the 1/10 and $1/10^2$ dilutions, and manual inspection revealed a single read at the $1/10^4$ dilution with identical

soft clips (Fig S5). The t(14;16) translocation in KMS11 was called by BreakDancer only at the 1/10 dilution but identified by manual inspection at 1/10, $1/10^2$, and $1/10^3$ dilutions (Fig S5).

In the RPMI-8226 cell line, two breakpoints were identified for t(16;22), and in MM1S, a single breakpoint was found for t(14;16). With genomic DNA sheared to 300 bp, these translocations were identified by BreakDancer (and manually verified) down to a $1/10^3$ dilution for one breakpoint and $1/10^2$ for the other breakpoint for t(16;22) in RPMI-8226. The limit of detection was $1/10^2$ for the sole t(14;16) breakpoint in MM1S.

For genomic DNA sheared to 300 bp, two clonal V(D)J rearrangements were identified for each of KMS11, RPMI-8226, and MM1S cell lines. In both dilution series, evidence of at least one rearrangement was detected down to a dilution of $1/10^3$ (Table 3).

For DNA sheared to 150 bp, translocation breakpoints were found down to 1% for all breakpoints in all cell lines, except for RPMI-8226, where one breakpoint was present down to 0.1% dilution, although at least one clonal V(D)J rearrangement in each dilution series was still identified down to $1/10^3$ (data not shown). This suggests that shorter fragment libraries may limit the ability of the method to detect low-frequency translocations, likely because of challenges mapping shorter sequences across breakpoints or differences in hybrid-capture dynamics when only a portion of the fragment matches the synthetic probe.

Known mutations *FGFR* p.Y373C in KMS11, *KRAS* p.G12 and *TP53* p.E285K in RPMI-8226, and *KRAS* p.G12A in MM1S were used as markers for the detection of somatic variants in the dilution series. Variant analysis was done both with and without the use of molecular barcoding techniques (Fig S6). Without barcoding techniques, variant detection was successful only down to a 10% dilution (Table 3); however, with molecular barcoding techniques, all variants were detected to a limit of 0.1% (Table 3).

### Application of combined V(D)J and IG translocation panels to patient samples

Targeted capture sequencing was used to detect immunoglobulin gene rearrangements and translocations in patient samples. Using the filtering algorithms established with cell line studies, 103 candidate rearrangements with corresponding CDR3 sequences were identified in 29 CD38[+] CD138[+] flow cytometry-sorted bone marrow samples with active myeloma. 39 of 103 were verified by manual review to be true rearrangements. The remaining candidate rearrangements were artifactual, highlighting the need for manual review following the automated filters. At least one candidate V(D)J with a unique CDR3 was verified in 26 of these 29 patient bone marrow samples. Only patients with serial samples are shown in Tables 3, S10, and S11. 11 V(D)J rearrangements were detected in seven of eight bone marrow samples as shown in Table 4. One patient sample did not have a detectable V(D)J rearrangement. Additional bone marrow samples were sequenced for MRD analysis (below).

Four of four *IGH* rearrangements were detected by targeted sequencing in patients with available clinical FISH testing in contemporaneous bone marrow (Table 4, Patients R and S; and

1 and 17. * Some bone marrow samples underwent conventional DNA library preparation. Bone marrow samples for minimal residual disease analysis underwent barcoded DNA library preparation.

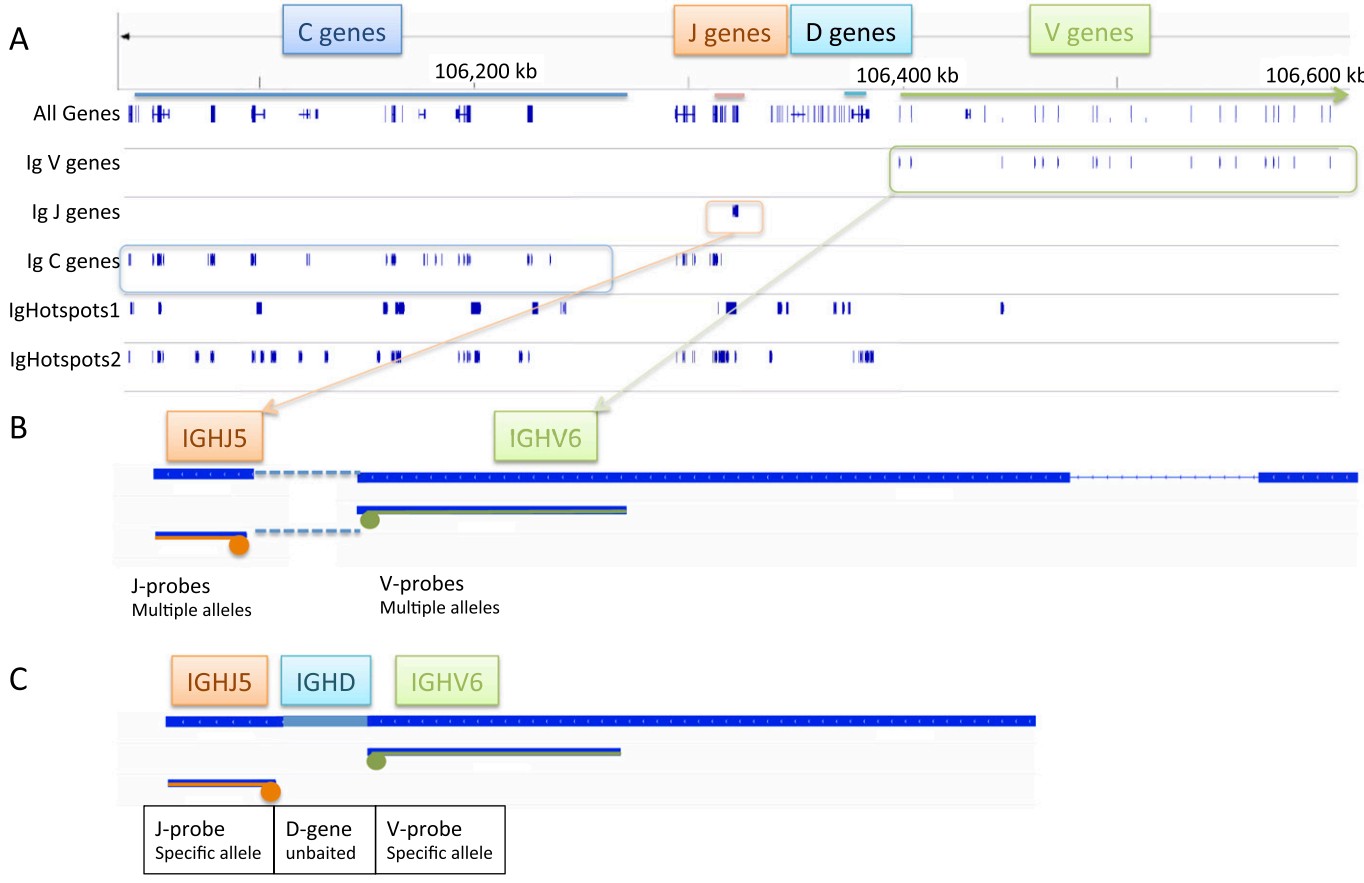

**Figure 2. Target Probe Design.**
**(A)** Target probes were designed in five different pools that can be combined together in a single capture, or used in any combination. The probe locations relative to the human genome are shown in a screenshot in the Integrative Genome Viewer[2]. IgHotspots1 pool was a part of a pre-existing probe set designed to target IGH translocation breakpoints and the *c-MYC* gene. IgHotspots2 included additional IGH translocation breakpoints and annotated hotspots close to or within *IGL* and *IGK* genes. Together, they are referred to as IgHotspots. **(B)** Immunoglobulin V-region genes were designed to probe the 3′ ends of the V gene, and J-region probes were designed to probe the 5′ ends of J genes in order to increase the likelihood of selecting fragments that cross the V(D)J junctions. Many alleles may overlap in the same genomic locations. **(C)** Captured fragment of rearranged V(D)J with specific probes. D regions are not baited because of their small size.

Table S10, Patients E and H). In another three samples, targeted sequencing testing detected IGH rearrangements with no available clinical FISH result (Patients F, I, and AB). In two patients, targeted sequencing detected a known *IGH* translocation in follow-up bone marrow samples with persistent/residual disease (Table 4, Patient S; and Table S4, Patients E and AA). Translocation breakpoints were specific to the patient samples.

Three serial bone marrow DNA samples were available from one multiple myeloma patient. Translocation t(11;14) and three V(D)J rearrangements were identified in the diagnostic sample that were present at decreasing read counts in the two follow-up samples. This corresponded to MFC counts of 0.03% and 0.02% abnormal plasma cells at 100 d and 6 mo post-transplant, respectively (Patient E, Table S10 and Fig S7).

### Application of targeted capture sequencing to patient cell-free DNA samples

To assess the ability of cell-free DNA sequencing to recapitulate IG rearrangements found in bone marrow, we captured and sequenced 39 contemporaneous or sequential cfDNA samples from 18 patients (Tables 4 and S10). cfDNA naturally occurs as 150- to 170-bp fragments and was input directly into DNA library preparation without shearing. 25 samples for MRD detection were sequenced to a depth of 8,000–10,000X coverage (Table S10), and 14 samples in patients with clinically measurable disease were sequenced to 2,000–5,000X coverage (Table 4).

### Patient bone marrow and cfDNA: combined V(D)J, IG translocation, and mutation detection

Bone marrow samples with contemporaneous or follow-up cfDNA samples were available for eight patients with clinically measurable disease (Table 2). For bone marrow samples, V(D)J rearrangements and immunoglobulin rearrangements were determined as described above and somatic mutations were previously determined from hybrid-capture experiments with the 38-gene panel alone. Four patients had translocations detected on bone marrow samples that were similarly detected in four of four contemporaneous cfDNA samples and two of four follow-up samples. Clinical FISH was

**Table 1.** Targeted Capture Sequencing (CapIG-seq) compared to standard PCR-based method (LymphoTrack) for identifying CDR3 of IGH rearrangements in myeloma cell lines.

| Cell line | CapIG-seq | | | | LymphoTrack | | | |
|---|---|---|---|---|---|---|---|---|
| | IGHV | IGHD | IGHJ | Clonal fraction | IGHV | IGHJ | Fraction total reads | CDR3 (common) |
| ALMC1 | IGHV3-21 | IGHD2-21 | IGHJ4 | 0.2857 | IGHV3-21 | IGHJ4 | 0.5090 | GTGAGAGCGTGGGGTGGGGAACTGTGG TGGTTACCAGGCTAC |
| | IGHV3-29 | IGHD3-3 | IGHJ6 | 0.2704 | Not found | Not found | | ACATAAGGTTCCAAGTGAGGAAACATC GGTGTGAGTCCAGACACAAA ATTTCCTGCAAAAAGAAGAAAGGAGTC[a] |
| EJM | IGHV2-5 | IGHD3-22 | IGHJ4 | 0.7765 | IGHV2-5 | IGHJ4 | 0.7065 | GCACACTTCCCCTCGCCTACCTCTGAT AATAATGGTTATTACTTTGACTAC |
| FR4 | IGHV3-7 | IGHD4-11 | IGHJ4 | 0.6575 | IGHV3-7 | IGHJ4 | 0.6618 | GCACGAGAGCAACTCAAAGGTACT GTAGTGGCTGCCCGGATGAC |
| H1112 | IGHV3-9 | IGHD2-8 | IGHJ3 | 0.4744 | IGHV3-9 | IGHJ3 | 0.5887 | GCAAGAGATAGCTCTATGGGGGGCGGA GACGACAATGGTCATCTTTTTGACATG |
| JJN3 | IGHV4-59 | IGHD1-26 | IGHJ4 | 0.9726 | IGHV4-59 | IGHJ4 | 0.6988 | GCGAAACCGTATAGTGGGAGCTACCCCG ACGGTCACTTTGGGCTAC |
| JMW1 | IGHV4-39 | IGHD3-10 | IGHJ5 | 0.0227 | IGHV4-39 | IGHJ5 | 0.2841 | GCGAGACACGTAAGGCAGGTCGGGGCC GACTGCTTCGACCCC[b] |
| | IGHV4-39 | IGHD3-10 | IGHJ5 | 0.7914 | IGHV4-39 | IGHJ5 | 0.2590 | GCGAGACATTTGAGGCAGGTCGGGG CCGACTGGTTCGACCCC |
| Karpas25 | IGHV4-4 | IGHD3-3 | IGHJ4 | 0.6384 | IGHV4-4 | IGHJ4 | 0.7129 | GCGAGAGAGACTGGGGGCGATTTCGATC GTTGGAGTGGCCAGCACTACTACTTTGACTCC |
| KP6 | IGHV3-33 | IGHD1-7 | IGHJ2 | 0.4906 | IGHV3-33 | IGHJ2 | 0.7018 | GCGAGAGAGTGGGAACTACGCTCGGG CTGGCACTTCGATCTC |
| LP1 | IGHV3-30 | IGHD2-8 | IGHJ6 | 0.6876 | IGHV3-30 | IGHJ6 | 0.6735 | GCGAAGACATTATTACAGATGGGGACAAGGG GCCACTACTACGGTTTGGACGTC |
| MM1S | IGHV3-30 | IGHD2-2 | IGHJ6 | 0.8802 | IGHV3-30 | IGHJ6 | 0.6864 | GCGAGAGATTTGAGAGGTTAGGGTGAA AGGTTCCTTGTTTGTAGTAGTACCAGCTGCTACG AGGACTCCTACTACTACGATATGGACGTC |
| OCI_MY5 | IGHV4-4 | IGHD1-26 | IGHJ4 | 0.5823 | IGHV4-4 | IGHJ4 | 0.7158 | GCGAGTGAGGGACAGGTGGGAAGTCAGGACTAC |
| SKMM1 | IGHV4-39 | IGHD3-22 | IGHJ4 | 0.4260 | IGHV4-39 | IGHJ4 | 0.6792 | GCGGGCATGGGAGTGGCGAGGCATAACTAT GATCATTGTGCTTCTTACTGGGTGGCCAC |
| XG2 | IGHV4-4 | IGHD6-19 | IGHJ6 | 0.6423 | IGHV4-61 | IGHJ6 | 0.6460 | GCGAGAATAGCCGTGGCTGGTAGTAGGGACTTTT ACAACTACAACCACGATATGGACGTC |

[a]CapIG-seq subsequently filtered out this sequence for lack of specificity.
[b]CapIG-seq did not include this sequence in the final output because of low clonal fraction.

available to confirm two of these translocations (Table 4). At least one V(D)J rearrangement was detected in seven of eight bone marrow samples. Detected V(D)J rearrangements were sample-specific and confirmed in five of seven contemporaneous and follow-up cfDNA samples. In the remaining two samples, one of two rearrangements was confirmed in one patient (Patient T) and zero of one rearrangement was confirmed in the other (Patient R); both samples had low amounts of cfDNA available for library preparation (Tables 4 and S11). One of these low DNA input follow-up samples (Patient R) may also account for a failure to detect a t(11;14) translocation, highlighting the need for adequate DNA input (Table S11).

Variant analysis was performed using a standard filtering algorithm as previously described, as barcoded libraries were not available for these samples. Two cfDNA samples (Patients U and V) were omitted from variant analysis for low coverage (<3,000X), leaving seven contemporaneous samples and five follow-up

samples for analysis. Most of the mutations detected on target capture sequencing of bone marrow samples were also detected in contemporaneous and follow-up cfDNA samples (22 of 33 mutations across 12 samples). In some cases, additional variants were detected in follow-up cfDNA samples. In all patients with longitudinal sampling, at least one class of genomic alteration was detected and available to track disease over time.

## Longitudinal testing and MRD

10 patients with multiple myeloma treated with autologous stem cell transplant had longitudinal bone marrow and cfDNA samples collected for sequential MFC and targeted capture sequencing for immunoglobulin translocations and V(D)J rearrangements (Table S10). To maximize sensitivity, most of the available cfDNA was used for library input (Table S11). In 2 of 10 cases (Patients A and F),

**Table 2.   Targeted Capture Sequencing (CapIG-seq) identification of recurrent translocations in myeloma cell lines.**

| Family/ Gene | Translocation | Cell lines carrying translocation | Algorithm called | Manual review | Sensitivity |
|---|---|---|---|---|---|
| c-MAF | t(14;16) | ANBL6, ARD, ARP1, CAG, JJN3, KMS11, KMS26, MM1R, MM1S, OCI-MY5, PCM6 | 10/11 | 10/11 | 0.91 (0.74–1.08) |
| | t(16;22) | XG6, Colo77, RPMI-8226 | 3/3 | 3/3 | 1.00 (1.00–1.00) |
| MAFB | t(14;20) | ALMC1, ALMC2, EJM, SKMM1 | 3/4 | 3/4 | 0.75 (0.35–1.15) |
| | t(8;20) | H929 | 1/1 | 1/1 | 1.00 (1.00–1.00) |
| | t(20;22) | L363 | 0/1 | 0/1 | 0.00 (0.00–0.00) |
| FGFR/ MMSET | t(4;14) | JIM1, JIM3, JMW1, KAS-6/1, KHM1, KMS11, KMS11adh, KMS11sus, KMS18, KMS26, KMS28BM, KMS28PE, KMS34, LP1, NCI-H929, OPM1, OPM2, PE2, UTMC2, XG7 | 18/20 | 20/20 | 1.00 (1.00–1.00) |
| CCND1 | t(11;14) | FLAM76, H1112, INA6, Karpas620, KMS12BM, KMS12PE, KMS21BM, KMS27, MOLP8, OCI-MY7, PE1, SKMM2, U266, XG1 | 7/14 | 10/14 | 0.71 (0.47–0.95) |
| CCND2 | t(12;14) | AMO1, XG2 | 1/2 | 1/2 | 0.50 (−0.19–1.19) |
| CCND3 | t(6;14) | SKMM1, FR4, KMM1 | 2/3 | 2/3 | 0.67 (0.14–1.2) |
| | t(6;22) | OCI-MY1 | 1/1 | 1/1 | 1.00 (1.00–1.00) |
| Novel translocations | | | | | |
| | t(14;15) | KMS20 | Y | Y | |
| | t(14;17) | KMS21BM | Y | Y | |
| | t(14;18) | ARP, ARP1, CAG | Y | Y | |
| *not c-MAF | t(14;16) | EJM | Y | Y | |
| *not-CCND1 | t(11;14) | L363 | Y | Y | |

patients did not have detectable disease either by MFC or by sequencing after undergoing autologous stem cell transplant. In 2 of 10 cases, the residual disease was present and detected by both clinical flow and at least one sequencing marker (Patients E and Z). For one patient (G), MFC detected MRD at three follow-up timepoints, with concordance by sequencing at two of three cases. In four cases (B, D, H, and AB), the residual disease was detected by MFC but not by a sequencing marker. In one case, t(11;14) was detected by sequencing in a follow-up sample where MFC was considered negative in a sample known to have this marker at diagnosis (Patient AA).

# Discussion

Targeted capture sequencing is a feasible, sensitive, and flexible strategy to detect recurrent primary translocations and V(D)J rearrangements in multiple myeloma and can be combined with variant detection in order to discover multiple genomic alterations in one assay. This strategy was successful in identifying most of the translocations that are currently detected in the clinic by FISH—100% of t(4;14), 86% of MAF family translocations, and 70% of cyclin D family translocations in myeloma cell lines. Unlike clinical FISH, this approach also has the ability to detect less common IGH partners, such as t(12;14) (*IGH-CCND2*) and t(14;20) (*IGH-MAFB*), and the potential to discover novel partners at these breakpoints. Other

groups have had success with detecting multiple genomic alterations in one assay (Yellapantula et al, 2019); however, we have also demonstrated the ability to include V(D)J testing and the application of the method to cfDNA. This further facilitates clinical testing by opening up a route to non-invasive clonotyping and monitoring of B-cell malignancies.

V(D)J rearrangements are a highly specific clonal marker widely used in MRD assays in hematologic malignancies, primarily by PCR-based techniques (Herrera & Armand, 2017). In our experiments, targeted capture and PCR-based assays were both successful in identifying *IGH* rearrangements in all cell lines tested. However, SHM is thought to be a primary limitation in PCR-based approaches in post-germinal center B cells because of frequent mismatches between consensus primers and germline sequence of rearranged and hypermutated *IGH* gene (Garcia-Sanz et al, 1999; González et al, 2003). Multiple myeloma, in particular, has been shown to have high SHM rates (9% on average, reported up to 23% in some series) (Garcia-Sanz et al, 1999). Targeted capture sequencing has the advantage of increased tolerance to SHM compared with PCR, which historically has a failure rate of ~10% (Gnirke et al, 2009). Increased tolerance with targeted capture sequencing has enabled polymorphism and genomic diversity studies that is not possible with PCR-based methods (Gasc et al, 2016), making targeted sequencing the ideal method to study the immunoglobulin repertoire.

Somatic variants, particularly disease-causing mutations, have been used for MRD detection, particularly when response to

**Table 3. Limit of detection of translocations, V(D)J rearrangements and mutations in myeloma cell lines diluted into peripheral blood mononuclear cells.**

| Cell line | Dilution | Translocations | | | | IG rearrangements | | Somatic mutations | | | |
| | | Breakpoint 1 | Breakpoint 2 | Breakpoint 3 | Breakpoint 4 | V(D)J 1 | V(D)J 2 | Barcoding analysis[a] | | Filtering analysis | |
| | | | | | | | | Mutation 1 | Mutation 2 | Mutation 1 | Mutation 2 |
| KMS11 | 1/10 | t(4;14)—1 | t(4;14)—2 | t(4;14)—3 | t(14;16) | IGKV3-15_J5 | IGKV1-37_J4 | FGFR3 p.Y373C | NA | FGFR3 p.Y373C | NA |
| | $1/10^2$ | t(4;14)—1 | t(4;14)—2 | t(4;14)—3 | t(14;16) | IGKV3-15_J5 | IGKV1-37_J4 | FGFR3 p.Y373C | | | |
| | $1/10^3$ | t(4;14)—1 | | | t(14;16) | IGKV3-15_J5 | | FGFR3 p.Y373C | | | |
| | $1/10^4$ | | | t(4;14)—3 | | | | | | | |
| | $1/10^5$ | t(4;14)—1 | | | | | | | | | |
| | $1/10^6$ | | | | | | | | | | |
| RPMI-8226 | 1/10 | t(16;22)—1 | t(16;22)—2 | NA | NA | IGKV2-28_J4 | IGLV2-14_J3 | KRAS p.G12A | TP53 p.E285K | KRAS p.G12A | TP53 p.E285K |
| | $1/10^2$ | t(16;22)—1 | t(16;22)—2 | | | IGKV2-28_J4 | IGLV2-14_J3 | KRAS p.G12A | TP53 p.E285K | | |
| | $1/10^3$ | t(16;22)—1 | | | | IGKV2-28_J4 | | KRAS p.G12A | TP53 p.E285K | | |
| | $1/10^4$ | | | | | | | | | | |
| | $1/10^5$ | | | | | | | | | | |
| | $1/10^6$ | | | | | | | | | | |
| MM1S | 1/10 | t(14;16) | NA | NA | NA | IGHV3-30_D2-2_J6 | IGLV2-14_J3 | KRAS p.G12A | NA | KRAS p.G12A | NA |
| | $1/10^2$ | t(14;16) | | | | IGHV3-30_D2-2_J6 | IGLV2-14_J3 | KRAS p.G12A | | | |
| | $1/10^3$ | | | | | IGHV3-30_D2-2_J6 | IGLV2-14_J3 | KRAS p.G12A | | | |
| | $1/10^4$ | | | | | | | | | | |
| | $1/10^5$ | | | | | | | | | | |
| | $1/10^6$ | | | | | | | | | | |

aAll four of these mutations were all detected down to $1/10^3$ with barcoding but filtered out after 1/10 with LOD score calling.

**Table 4. CapIG-Seq detection of multiple genomic alterations in longitudinal/contemporaneous bone marrow and cfDNA in myeloma patients.**

| Patient | Genomic alteration/marker | Clinical bone marrow FISH | | | Bone marrow sequencing | cfDNA contemporaneous samples | cfDNA follow-up 1 samples | cfDNA follow-up 2 samples |
|---|---|---|---|---|---|---|---|---|
| | | t(4;14) | t(11;14) | t(14;16) | | | | |
| Patient R | t(11;14) | Negative | Detected | Negative | Detected | Detected | | |
| | IGHV3-66/D3-22/J5 | | | | Detected (0.69) | | | |
| | IGLV2-23/J2 | | | | Detected (0.8) | | | |
| | NRAS p.Q61K | | | | Detected (0.58) | Detected (0.016) | | |
| | FAM46C p.F274L | | | | Detected (0.46) | Detected (0.0062) | | |
| | LTB p.P75L | | | | Detected (0.50) | | Detected (0.43) | |
| Patient S | t(4;14) | Detected | Negative | Negative | Detected | Detected | Detected | |
| | IGKV1-37/J3 | | | | Detected (0.61) | Detected (0.61) | Detected (0.82) | |
| | IGKV1-5/J4 | | | | Detected (0.11) | Detected (0.11) | Detected (0.12) | |
| | KRAS p.A146T | | | | Detected (0.19) | | Detected (0.065) | |
| | FAM46C p.D150Y | | | | Detected (0.23) | | | |
| | PRKD2 p.Y566C | | | | Detected (0.65) | PRKD2 p.Y566C (0.040) | Detected (0.11) | |
| Patient T | No translocations detected | Negative | | | No translocations detected | | | |
| | IGKV3-11/J3 | | | | Detected (0.62) | Detected (0.55) | | |
| | IGHV2-5/D3-3/J6 | | | | Detected (0.21) | | | |
| | KRAS p.Q61H | | | | Detected (0.047) | Detected (0.015) | | |
| | PRDM1 p.P66S | | | | Detected (0.39) | Detected (0.46) | | |
| Patient U | t(12;14) | Negative | | | t(12;14) | t(12;14) | t(12;14) | |
| | IGKV4-1/J2 | | | | Detected (0.59) | Detected (0.04) | Detected (0.21) | |
| | MAX p.R27W | | | | Detected (0.90) | Detected (0.54) | Detected (0.65) | Sample failed for mutation calls |
| | KRAS p.G13D | | | | Detected (0.080) | Detected (0.19) | Detected (0.091) | |
| | CYLD p.G930G | | | | | | Detected (0.017) | |
| Patient V | t(12;14) | Negative | | | t(12;14) | t(12;14) | | |
| | IGKV1-39/J1 | | | | Detected (0.68) | Detected (0.70) | | |
| | IGKV4-1/J4 | | | | Detected (0.29) | Detected (0.29) | | |
| | KRAS p.G12V | | | | Detected (0.45) | Sample fail for mutations | | |
| | BRAF p.D594N | | | | Detected (0.45) | | | |
| Patient W | No translocations detected | Negative | | | No translocations detected | No translocations detected | | |
| | No V(D)J rearrangement found | | | | | | | |
| | IDH1 p.F86I | | | | Detected (0.091) | Detected (0.085) | Detected (0.089) | |
| | ATR p.E650K | | | | Detected (0.089) | Detected (0.094) | Detected (0.089) | |

**Table 4.** Continued

| Patient | Genomic alteration/marker | Clinical bone marrow FISH | | | Bone marrow sequencing | cFDNA contemporaneous samples | cFDNA follow-up 1 samples | cFDNA follow-up 2 samples |
|---|---|---|---|---|---|---|---|---|
| | | t(4;14) | t(11;14) | t(14;16) | | | | |
| | PIK3CA p.H59P | | | | Detected (0.41) | Detected (0.38) | Detected (0.37) | |
| | CYLD p.L135M | | | | | Detected (0.044) | | |
| | CYLD p.G930G | | | | | Detected (0.045) | | |
| | IKZF3 p.S88R | | | | | Detected (0.039) | | |
| Patient X | No translocations detected | Negative | | | No translocations detected | No translocations detected | | |
| | IGLV1-44/J2 | | | | Detected (0.83) | Detected (0.0069) | Detected (0.0099) | |
| | NRAS p.G13R (0.35) | | | | Detected (0.35) | | Detected (0.73) | |
| | FAM46C p.G37V (0.25) | | | | Detected (0.25) | | | |
| | MAX p.R51W (0.72) | | | | Detected (0.72) | | | |
| Patient Y | No translocations detected | Negative | | | No translocations detected | No translocations detected | | |
| | IGKV3-20/J2 | | | | Detected (0.55) | Detected (0.54) | | |
| | EGR1 p.S42T | | | | Detected (0.41) | Detected (0.34) | | |
| | KRAS p.G13D | | | | Detected (0.79) | Detected (0.48) | | |
| | ZFHX4 p.P3167P | | | | | Detected (0.014) | | |

☐ Concordance with bone marrow/clinical data
☐ Found in bone marrow but not in cfDNA
☐ Found in cfDNA and not in bone marrow
☐ Not applicable/no sample.

targeted therapy is desired (Trudel et al, 2016). In myeloma, therapeutic selection pressures have been demonstrated to result in the expansion of minor subclones and "clonal tiding." It is thus especially advantageous to monitor multiple variants and multiple genomic alterations simultaneously. We have previously demonstrated the ability to follow patients treated with targeted agents over time (Trudel et al, 2016) and to replicate variants detected in bone marrow with contemporaneous cfDNA samples (Kis et al, 2017b). Variant detection is also greatly improved with barcoding techniques as exemplified in cell line and limit-of-dilution experiments. Relatively lower coverage in the current work may explain decreased sensitivity in patient samples where barcoded DNA libraries were not available. Pairing mutation testing with unique translocation and V(D)J detection both improves confidence in the detected genomic alternations and enables improved tracking of subclones over time.

Structural rearrangements, including translocations and V(D)J rearrangements, present at even low allelic fractions may be detected in sequencing data with a high degree of confidence, particularly if a purified tumor sample is available for reference. The identification of structural rearrangements and V(D)J rearrangements is highly specific to the tumor sample and is unlikely to be a result of sequencing artifact. This is a significant advantage over techniques that rely on variant detection for MRD detection. This approach may also be used for the detection of uncommon translocations and the peripheral blood detection of translocations alongside usual clinical biomarkers of disease.

Several limitations and opportunities for development arise from this work. Although we were able to detect evidence of translocations to a sensitivity of $1/10^5$ DNA fragments in the KMS11 cell line dilution series with DNA fragments sheared to 300 bp, this was not reproducible in different cell lines or with some patient samples. This may be because of the amount of input DNA required to reliably detect genomic alterations. 500 ng of DNA was used as input for DNA libraries, translating to ~83,000 normal genomic equivalents (less in hyperdiploid myeloma cell lines), which may not have reached the required input for sensitive MRD detection. These limit-of-detection findings were also not reproducible with shorter 150-bp DNA fragments. A minimal length spanning the junction between structural rearrangements may be required to detect such structural changes. Previous work with T-cell receptor V(D)J rearrangements has estimated >99% sensitivity to detect V-J rearrangements with a fragment length of 182 bp (Mahé, 2016).

In spite of this, cfDNA and bone marrow findings were highly concordant for V(D)J rearrangements in patients that had clinically detectable disease (Table 2), provided a minimum input of 83 ng of cfDNA for DNA library preparation. We were not as successful in post-transplant patients with MRD (Table S10). In addition to the structural limitations described, this may also be because of significantly decreased ctDNA shedding in cases where myeloma is relatively quiescent. Therefore, the detection of V(D)J rearrangements may be limited to use in either genomic DNA or where there is sufficient disease activity to generate the minimum required ctDNA.

Additional limitations with targeted sequencing for translocation detection include the requirement for knowledge of relatively precise breakpoints and the difficulties with repetitive or unmapped regions. For example, the known t(14;16) in the EJM cell line was not detected because of a missed hotspot location during probe design. The current probe set was also not designed to detect complex secondary translocations, including those with *c-MYC*, which often involve non-immunoglobulin partners (Affer et al, 2014). Although the *c-MYC* gene and some *IGH* breakpoints were baited (Affer et al, 2014), many rearrangement breakpoints that result in translocations of *c-MYC* were not captured in the probe design. Thus, the primary t(11;14) event is detectable in the Karpas620 cell line, but not the additional (secondary) t(8;14). Repetitive and unmapped regions of the genome near the chromosome 11 breakpoint in KMS12 resulted in difficulties with alignment and translocation calling, potentially accounting for the inability to detect a proportion of t(11;14) translocations. We were nonetheless able to identify most of the clinically tested translocations, and a modular design allows for flexibility in adding or combining probe sets as necessary, with additional knowledge or depending on the specific clinical application.

The ability to detect primary translocations and V(D)J rearrangements in multiple myeloma by targeted sequencing has multiple potential applications. Current technologies primarily use FISH for translocation detection, MFC or PCR-based molecular assays for MRD detection, and targeted sequencing for mutation detection in multiple myeloma. In contrast, this approach has the potential to detect all genetic alterations in a single assay, conserving sample material and minimizing resources. In its current form, CapIG-seq is complementary to clinical FISH, with the ability to detect additional rearrangements and mutations that are not present in standard-of-care testing. The addition of CapIG-seq to upfront testing in all comers may yield valuable information to guide clinical care. One advantage of CapIG-seq is its flexibility to modify testing in patient samples at follow-up timepoints. In clinical practice, this may take the form of testing for specific mutations to determine eligibility for clinical trials or novel agents, or testing for specific genomic alterations that give rise to therapeutic resistance. Although cfDNA testing is challenged by low concentrations of ctDNA at follow-up timepoints, assay sensitivity can be improved by increasing the number of genetic changes identified with a CapIG-seq approach.

# Materials and Methods

Some methods describing the design and testing workflow have been presented in the sections above for improved readability.

### Materials

Genomic DNA from 65 commercially available multiple myeloma cell lines was donated by Dr. Jonathan Keats' Laboratory (Translational Genomics Research Institute). 24 cell lines (23 myeloma and one B-lymphoblast) from 16 donors had internal and publically available (Yang et al, 2014) WGS data. Seven cell lines were derived from three donors, representing biological replicates. Patients with multiple myeloma consented to the collection and use of their bone marrow and cell-free DNA samples from blood plasma for research purposes.

## Methods

### Barcoded DNA library preparation

Barcoded DNA libraries were prepared with KAPA Hyper Prep Kits for Illumina TruSeq library construction (Kapa Biosystems, Inc.) in conjunction with custom-designed double-stranded duplex molecular barcodes (duplexes) and index primers (indexes) (xGen Lockdown Custom Probes Mini Pool; Integrated DNA Technologies), designed in collaboration with Dr. Scott Bratman.

Genomic DNA was sheared to 150 or 300 bp in Tris-EDTA buffer using either the Covaris LE220 or the Covaris M220 sonicator (Covaris). End repair and A-tailing was performed according to the manufacturer's instructions. cfDNA did not require shearing and proceeded directly from extraction to end repair and A-tailing.

Custom-designed duplex barcodes (Table S12) were ligated to all samples overnight (12–16 h) at 4°C according to the manufacturer's protocol, using adapter: insert molar ratios ranging from 10:1 to 200:1 according to the manufacturer's protocol.

Following ligation, samples were cleaned up with 0.8× AMPure beads pre-equilibrated to room temperature for 30 min according to the manufacturer's protocol. Beads were incubated for 15 min to bind DNA to beads, and bead washes using the DynaMag-2 Magnetic tube rack (Thermo Fisher Scientific) were performed using 80% ethanol. Samples were eluted in 22 $\mu$l TE buffer with a 5-min room temperature incubation. PCR amplification was performed with sample-specific indexes and a universal primer and a minimum number of amplification cycles depending on the amount of input DNA. Samples were cleaned up again with 1× AMPure bead wash using 80% ethanol for washes and eluted in 30–50 $\mu$l of nuclease-free water for target capture.

Barcoded library preparation differs from conventional (non-barcoded) library preparation as illustrated in Fig S5.

### Gene panel probe design

**Probe design for V(D)J detection** We designed a targeted IG capture panel using single-stranded, 120 nucleotide synthetic probes for target enrichment (xGen Lockdown Custom Probes Mini Pool; Integrated DNA Technologies). The ImmunoGeneTics (IMGT) reference database (Giudicelli et al, 2005; Lefranc, 2017) annotates known variants of immunoglobulin V, D, and J genes. Similar to a strategy we employed for T-cell receptor sequencing (Mulder et al, 2018), we designed probes to hybridize to the 3′ ends of all IMGT-annotated V genes and the 5′ ends of J genes to maximize the likelihood of capturing DNA fragments spanning the V-J or V(D)J rearrangement junction and incorporated barcoded library preparation to improve variant calling (Fig 2).

596 individual V-region sequences were extracted from the IMGT database and truncated at 120 bases from the 3′ end of the V gene. Single-stranded, biotinylated synthetic probes were synthesized (xGen Lockdown Custom Probes Mini Pool; Integrated DNA Technologies). Initial target capture experiments with this probe set resulted in a high proportion of off-target sequencing reads (>97%). After optimization by removing eight probes predicted to have a high degree of promiscuity and shifting in genomic position (with an alignment to human genome reference hg19) of two other probes, a second version comprising 588 V-region probes was synthesized. The 588 V-region probe set yielded an off-target rate of 20–30% and was used for this work (Table S2).

57 J-region sequences were extracted from the IMGT database, and 118 C-region sequences were collated from extracted sequences from IMGT and sequences from exons as annotated in the reference human genome hg19. J-region and C-region panels did not require additional optimization (Table S2).

**Probe design for translocation detection** Probes designed to hybridize to regions containing translocation hotspots within the immunoglobulin genes were designed to detect primary rearrangements in multiple myeloma and obtained from a literature review of known breakpoint regions where possible (Walker et al, 2013; Affer et al, 2014; Bolli et al, 2016). Constant (C gene) probes were also baited (described above), because errors within CSR within these regions are a known mechanism of illegitimate rearrangement in myeloma (Walker et al, 2013).

The IgHotspots1 panel is a 172-probe pool with targets within the *IGH* locus that was previously designed as a part of a five-panel set to study multiple genomic aspects of multiple myeloma (Kis et al, 2017a). The IgHotspots2 panel is a 200-probe pool designed with additional references to capture additional *IGH* translocations missed by IgHotspots1, and with targets within the *IGK* and *IGL* loci to capture primary light chain translocations. All five V(D)J and translocation-directed pools were combined for targeted capture sequencing of 65 cell lines.

**Probe design for somatic variant detection** A probe set directed against the exons of 38 genes with clinical relevance in multiple myeloma was used to detect somatic variants (Fig 1). The development and validation of somatic variant detection with both genomic and cfDNA in multiple myeloma is described in detail elsewhere (Kis et al, 2017b). The gene pools were designed and tested as a part of a previously described five-panel set to study multiple genomic aspects of myeloma (Kis et al, 2017a). Variant detection probe sets were combined with immunoglobulin and translocation probes for limit-of-detection experiments in cell lines and some patient samples as described below.

### Sequencing

Cell lines were sequenced on the Illumina NextSeq 500 with the 150-bp paired-end application to achieve 2,500–3,000X coverage. For limit-of-detection experiments using cell lines diluted into PBMCs, samples were sequenced on Illumina NextSeq 500 and HiSeq 2500 sequencers using a 150-bp paired-end application and achieving a combined depth of 120,00X.

Patient bone marrow samples were sheared to 300 bp and sequenced on the Illumina NextSeq 500, achieving 5,000X coverage. cfDNA samples from the blood plasma of 18 patients were available for sequencing (Tables 4 and S4). cfDNA naturally occurs as 150- to 170-bp fragments and was input directly into DNA library preparation without shearing. The 25 patient samples collected for MRD detection were sequenced to a depth of 8,000–10,000X coverage (Table S10), and the 14 samples in patients with clinically measurable disease were sequenced to 2,000–5,000X coverage (Table 4) using a 150-bp paired-end application on the Illumina NextSeq 500.

### Comparison between CapIG-seq and PCR-based testing (LymphoTrack assay)

We also the compared CapIG-seq assay with a PCR-based assay, LymphoTrack, for *IGH*V rearrangements and CDR3 sequences identified in 13 cell lines known to have IGHV-rearranged alleles. Genomic DNA was extracted with the Maxwell 16 FFPE Plus LEV DNA purification kit (Promega). For *IGH* gene sequencing, the Lympho-Track *IGHV* Leader Somatic Hypermutation Assay Panel was used according to the manufacturer's instructions and sequenced on the MiSeq (Illumina). The *IGH* sequences obtained were analyzed using IMGT program version: 3.5.24 (March 9, 2021). Four cell lines were run in duplicate at 100 ng (to compare with the same amount required for CapIG-seq) and 250 ng (manufacturer's protocol for Lympho-Track) with identical results (Table S5). Multiple *IGH* sequences at low frequency with the same CDR3 were collapsed into a single clonal fraction to compare with the CapIG-seq method (Tables 1 and S6).

### Bioinformatics tools and methods

**V(D)J rearrangement calling** MiXCR (Bolotin et al, 2015) exports candidate CDR3 sequences along with candidate immunoglobulin gene alleles and rearrangements for manual review. After removing pseudogenes, candidate rearrangements were manually reviewed (below) to develop an appropriate filtering algorithm for true rearrangements. From targeted sequencing of the 20 cell lines with available WGS data, 868 V(D)J rearrangements were called by MiXCR. Of these, 111 were manually reviewed.

**MiXCR filtering by clonal fraction** Clonal fractions were calculated as a percentage of total rearranged clones at the specific immunoglobulin locus. All true rearrangements had a clonal fraction of 0.15 or higher. False rearrangements were present in clonal fractions ranging from 0.005 to 0.5 of those examined. There were 25 falsely called rearrangements with a clonal fraction of 0.15 or higher; there were 38 falsely called rearrangements with a clonal fraction of 0.10 or higher. Drawing a threshold of 0.10 gave a sensitivity of 100% for detecting real V(D)J rearrangements and a specificity of 54% based on the manually reviewed rearrangements (Fig S2A). The specificity may be improved with further manual review to verify falsely called rearrangements; however, a higher sensitivity was preferred for the initial development of the filtering algorithm.

**MiXCR filtering by specificity** Each of the 868 clonal sequences with candidate V(D)J genes was compared with the remaining 867 to determine whether the identified sequence was identical to that found in any other cell lines in order to filter out recurrent sequence polymorphisms. The candidate V, D, or J alleles could not be used for this purpose as (1) SHM may result in slightly different sequences from the same candidate allele and (2) more than one candidate allele is nominated by MiXCR in the case of highly homologous genes. After filtering for a clonal fraction of >0.10, 128 candidate rearrangement sequences remained. Based on the 111 rearrangements that were manually reviewed, a threshold of five or fewer representations in the total repertoire of called rearrangements gave a sensitivity of 100% for detecting a real rearrangement (Fig S2B). Although this is confounded by multiple (up to 3) cell lines

deriving from the same donor within the subset of 19, a threshold of 5 was still required to achieve 100% sensitivity after removing duplicate cell lines from consideration (data not shown).

**V(D)J MiXCR filtering by absolute clone count** Filtering based on absolute clone count was performed following a similar principle of maximizing sensitivity for detecting true rearrangements. After filtering the initial 868 MiXCR calls for clonal fraction >0.10, a threshold of absolute clone count >50 gave 100% specificity and 100% sensitivity based on reviewed rearrangements (Fig S2C). For patient samples, a threshold of 10 was used for improved sensitivity.

### Translocation calling

Translocation calling was performed with both an in-house algorithm CluMP (clustering of mate pairs) and BreakDancer-Max (Chen et al, 2009). CluMP uses a combination of large insert size between paired-end reads and soft-clipped (misaligned) bases at the same sites to call candidate breakpoints. CluMP initially extracts read pairs aligned to different chromosomes or with large insert sizes, and those supporting a similar structural rearrangement are clustered together. Genomic coordinates of these clusters are used to identify candidate breakpoints. The algorithm next extracts high base-quality (Phred score 20 by default) and high mapping quality (Q 30) single reads within the defined breakpoint regions with a significant proportion (33%) of bases within the read that do not match reference sequence. For the current purpose, the base quality, the mapping quality, and the number of mismatched nucleotides were used at the default settings described. The minimum insert size was set to 5,000 kb. Candidate translocations called by CluMP were subjected to further filtering using the R statistical software (version 0.99.489; RStudio, Inc.) to remove repetitive regions (Table S13) and translocations within immunoglobulin regions that were instead analyzed by the MiXCR algorithm.

BreakDancer-Max similarly extracts aligned anomalous read pairs with large inserts to estimate breakpoints (Chen et al, 2009). BreakDancer-Max (version 1.1.2) was run to detect only inter-chromosomal translocations with a minimum MAPQ (MAPping Quality) filter of Q30 and coefficient of variation 10 on all samples. Intra- and inter-gene rearrangements within *IGL*, *IGK*, and *IGH* were filtered out. The R statistical software was used to filter out recurrent repetitive regions of the genome and regions with high sequence homology (Table S13) determined by manual inspection. Such regions were determined by visual inspection of called translocations occurring at high frequency in multiple cell lines.

All structural variants called by both CluMP and BreakDancer-Max were manually verified in IGV. Manual inspection at known breakpoints in cell lines was performed for all known translocations missed by either or both callers.

### Manual review for translocation detection

A rearrangement was deemed to be present based on a combination of large-insert read pairs and concordance of non-duplicated reads that do not align to the reference genome but map to the read pair (Fig S2A).

Large-insert read pairs map distant to each other on the same chromosome or on different chromosomes altogether. Clusters of large-insert read pairs that map to the same location suggest a structural rearrangement. This may be confounded by two homologous regions, which are filtered out by bioinformatics methods (Table S13).

Clusters of reads sharing the same genomic transition point from aligned to misaligned reads indicate a structural rearrangement breakpoint (i.e., transition from gray bar to multicolored reads in Fig S2A and B). Staggered ends of misaligned reads indicate unique reads, whereas PCR duplicates may generate a stack of reads with the same start and end locations. A translocation breakpoint, especially at the unbaited partner breakpoint (i.e., chr4 in t(4;14) translocation where target probes are located only on chr14), is also accompanied by a drop in coverage at the breakpoint.

In the IGV (Thorvaldsdottir et al, 2013), sorting sequencing reads by insert size clusters the read pairs and a reasonable proportion of misaligned single reads together, and coloring misalignments by nucleotide allows for manual inspection as to the concordant sequences of these reads.

In individual samples, a rearrangement or translocation was considered confirmed if there were at least two non-duplicate reads with the same misaligned sequences at both translocation and rearrangement partners. In samples with paired targeted sequencing or WGS data, one read at each partner was sufficient if the misaligned sequences were concordant with those of the paired sample and at least one of either the targeted sequencing or WGS sample had at least five non-duplicate misaligned reads (Fig S2).

### Limit of detection

We next combined V(D)J and translocation hotspot panels with panels with our previously developed somatic variant detection panel. We sought to determine the limit of detection with this approach using DNA from myeloma cell lines (KMS11, RPMI-8226, and MM1S) combined with DNA from PBMCs of one healthy volunteer. DNA was sheared to either 300 or 150 bp to mimic conventional sequencing libraries or cell-free DNA. Concentrations were targeted at six logarithmic intervals from 1/10 to $1/10^6$ dilutions of cell line into PBMC DNA. To maximize sensitivity, we used 500 ng of total input DNA for DNA library preparation and 500–800 ng of DNA library for target capture. Molecular barcoding techniques were used in library preparation to improve confidence in variant calling at lower allelic frequencies. DNA input for capture experiments was balanced for clusters generated per ng of DNA. Samples were sequenced on Illumina NextSeq 500 and HiSeq 2500 sequencers using a 150-bp paired-end application and achieving a combined depth of 120,00X for limit-of-detection experiments. V(D)J rearrangement and translocation calling methods were performed with MiXCR and BreakDancer or CluMP, respectively, as described above.

For the barcoded analysis, error correction was first achieved using molecular adapters to form consensus sequences, described elsewhere (Wang et al, 2019). Candidate mutations were then called using muTect (version 1.1.4) configured to allow the detection of rare variants without a matching normal (Kis et al, 2017b). Annotation was performed by Oncotator (version 1.2.8.0). For the standard filtering analysis, variant calls were filtered based on modified Z-scores derived from the tumor logarithm of odds (LOD) scores, defined as log (likelihood event is real/likelihood event is sequencing error) as previously described (Kis et al, 2017b).

## Data Availability

All raw and processed sequencing data generated in this study have been submitted to the European Nucleotide Archive at EMBL-EBI (https://www.ebi.ac.uk/) under accession number PRJEB48836.

## Supplementary Information

## Acknowledgements

This work was funded by the Canadian Cancer Society, Princess Margaret Cancer Foundation, and the Princess Margaret Cancer Centre Innovation Accelerator Fund. TJ Pugh holds the Canada Research Chair in Translational Genomics and is supported by a Senior Investigator Award from the Ontario Institute for Cancer Research and the Gattuso-Slaight Personalized Cancer Medicine Fund. We thank the staff of the Princess Margaret Genomics Centre (www.pmgenomics.ca, Troy Ketela) and Bioinformatics and HPC Core (Zhibin Lu) for their expertize in generating the sequencing data used in this study.

## Author Contributions

S Chow: conceptualization, data curation, formal analysis, validation, investigation, and methodology.
O Kis: conceptualization, data curation, formal analysis, investigation, and methodology.
DT Mulder: conceptualization, data curation, investigation, and methodology.
A Danesh: formal analysis and investigation.
J Bruce: formal analysis and investigation.
TT Wang: data curation and methodology.
D Reece: resources and funding acquisition.
N Bhalis: resources and investigation.
P Neri: resources and investigation.
PJB Sabatini: resources, investigation, and methodology.
J Keats: conceptualization, resources, and investigation.
S Trudel: conceptualization, data curation, supervision, investigation, and methodology.
TJ Pugh: conceptualization, data curation, formal analysis, supervision, funding acquisition, investigation, methodology, and writing—review and editing.

## Conflict of Interest Statement

The described method is the subject of a patent filing by the University Health Network "Hybrid-capture sequencing for determining immune cell clonality": Patent Application No. US16/093,825 (US11,149,312—issued); CA3,020,814 filed April 13, 2017; EP17781661.8 filed April 13, 2017; CA3,064,312, filed May 29, 2018; US16/617,826 filed May 29, 2018; and EP18810749.4 filed May 29, 2018. The remaining authors declare no conflicts of interest.

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
