## [Reviewer comments · Life Science Alliance]

Life Science Alliance

Myeloma Immunoglobulin Rearrangement & Translocation detection through Targeted Capture Sequencing

Signy Chow, Olena Kis, David Mulder, Arnavaz Danesh, Jeffrey Bruce, Ting Wang, Donna Reece, Nizar Bahlis, Paola Neri, Peter Sabatini, Jonathan Keats, Suzanne Trudel, and Trevor Pugh

DOI: <https://doi.org/10.26508/lsa.202201543>

Corresponding author(s): Trevor Pugh, Princess Margaret Cancer Centre and Signy Chow, University of Toronto

Review Timeline:

Submission Date:	2022-05-31
Editorial Decision:	2022-07-06
Revision Received:	2022-08-13
Editorial Decision:	2022-09-02
Revision Received:	2022-09-29
Accepted:	2022-09-30

Scientific Editor: Novella Guidi

Transaction Report:

July 6, 2022

Re: Life Science Alliance manuscript #LSA-2022-01543-T

Dr. Trevor J Pugh
Princess Margaret Cancer Centre, University Health Network
Toronto, ON
Canada

Dear Dr. Pugh,

Thank you for submitting your manuscript entitled "Immunoglobulin Rearrangement & Oncogenic Translocation detection in Multiple Myeloma through Targeted Capture Sequencing" to Life Science Alliance. The manuscript was assessed by expert reviewers, whose comments are appended to this letter. We invite you to submit a revised manuscript addressing the Reviewer comments.

Thank you for this interesting contribution to Life Science Alliance. We are looking forward to receiving your revised manuscript.

Sincerely,

B. MANUSCRIPT ORGANIZATION AND FORMATTING:

Reviewer #1 (Comments to the Authors (Required)):

In the article "Immunoglobulin Rearrangement & Oncogenic Translocation detection in Multiple Myeloma through Targeted Capture Sequencing" [LSA-2022-01543-T], Chow, Pugh, and colleagues presented a novel hybrid-capture approach (CapIG-seq) for the detection of immunoglobulin V(D)J rearrangements, oncogenic translocations, and gene mutations. The assay was tested on both model systems and patient specimens and was compared with PCR-based sequencing techniques as well as FISH. Longitudinal analyses for the same patients were performed and non-invasive disease monitoring was explored. Data presentation for each point was clear and thorough and demonstrated the utility of the assay. Genetic follow-up of plasma cell neoplasm (PCN) is often challenged by the limited number of cells available. The study presents a practical approach to workup PCN clinically that characterizes structural rearrangements and gene mutations in the same assay.

1. Although this is a method-focused paper, the reviewer suggests that the manuscript expands on exploring scenarios as to how this assay would be used in the context of routine tests such as FISH. What would be the added cost and turn-around-time (TAT) if applicable? Would the authors suggest this as an assay for all comers or would a selective and reflexive approach be more appropriate?

2. Minor comment: in table 1, MM1S IGHJ under LymphoTrack is missing a number.

Reviewer #2 (Comments to the Authors (Required)):

Multiple myeloma is a hematological disease characterized by unstable genome, various translocations and clonal immunoglobulin variations. This paper describes a novel hybrid-capture approach targeting both ends of V and J segments of immunoglobulins to detect VDJ rearrangements as well as oncogenic translocations.

Data are strongly supportive. The manuscript is clear. No additional experiments required in my opinion.

Reviewer #3 (Comments to the Authors (Required)):

In this paper, Chow et al present their work on the development of a sensitive capture based method to detect V(D)J rearrangements, IGH translocations and mutations in MM samples.

The work is mainly technical, and packed with details. This is to be praised, even if some issues arise from disconnects between tables and text in terms of numbers, and should be fixed because they make the results very hard to follow.

Also, while their technical approach is somehow different from others, limitations in cfDNA have been extensively described (Oberle et al 2017, Manzoni et al 2020), and potential advantages of an "all in one" approaches have been published as well by several groups.

Therefore, the novelty is relatively low, even if there is some confirmatory value

- One important message is that shorter DNA fragments -150bp- may carry limited value to map IGH Tx breakpoints. The authors use cellular DNA to infer implications for cfDNA usage. However, aside from size, also quality and quantity of cfDNA can be a problem, and I would be careful translating findings here to the cfDNA field. It could be quite worse than this.

- In table S4A, there are some discrepancies in numbers: are there 48 sequences (as in the table) or 47 (as in the text)? Similarly, WGS algorithm called 32 or 30?

Reviewer #1

In the article "Immunoglobulin Rearrangement & Oncogenic Translocation detection in Multiple Myeloma through Targeted Capture Sequencing" [LSA-2022-01543-T], Chow, Pugh, and colleagues presented a novel hybrid-capture approach (CapIG-seq) for the detection of immunoglobulin V(D)J rearrangements, oncogenic translocations, and gene mutations. The assay was tested on both model systems and patient specimens and was compared with PCR-based sequencing techniques as well as FISH. Longitudinal analyses for the same patients were performed and non-invasive disease monitoring was explored. Data presentation for each point was clear and thorough and demonstrated the utility of the assay. Genetic follow-up of plasma cell neoplasm (PCN) is often challenged by the limited number of cells available. The study presents a practical approach to workup PCN clinically that characterizes structural rearrangements and gene mutations in the same assay.

1. Although this is a method-focused paper, the reviewer suggests that the manuscript expands on exploring scenarios as to how this assay would be used in the context of routine tests such as FISH. What would be the added cost and turn-around-time (TAT) if applicable? Would the authors suggest this as an assay for all comers or would a selective and reflexive approach be more appropriate?

Thank you for this suggestion. We think that there could be multiple clinical and research applications for this assay. In its current form, CapIG-seq would be complementary to clinical FISH and cytogenetic assays. We would suggest using CapIG-Seq at diagnosis for all newly diagnosed patients and tailoring the sequencing panel to individuals at follow up time points if laboratory workflow allows. Overall, we would suggest a broader strategy at diagnosis and more targeted strategy at follow up sampling, with the ability to re-expand testing at the time progression or suspected therapeutic resistance. Due to the rapidly falling costs of high throughput sequencing as well as variations in adoption by different laboratories, it is difficult to comment precisely on the cost or turn around time (TAT) of this assay.

However, we anticipate that the cost would be similar to other sequencing panels of the same size used in clinical practice and it is currently equivalent to the cost of a single FISH assay. TAT may range from 1-4 weeks depending on institutional adaptation. In our laboratory workflow, the assay can be done in 1-2 weeks, including analysis and review, not accounting for wait times for sequencer availability. We have elaborated on some of these suggestions in the discussion, excerpted below:

In its current form, CapIG-seq is complementary to clinical FISH, with the ability to detect additional rearrangements and mutations that are not present in standard of care testing. The addition of CapIG-seq to upfront testing in all comers may yield valuable information to guide clinical care. One advantage of CapIG-seq is its flexibility to modify testing in patient samples at follow-up timepoints. In clinical practice, this may take the form of testing for specific mutations to determine eligibility for clinical trials or novel agents, or testing for specific genomic alterations that give rise to therapeutic resistance. Although cfDNA testing is challenged by low concentrations of ctDNA at follow up time-points, assay sensitivity can be improved by increasing the number of genetic changes identified with a CapIG-seq approach.

2. Minor comment: in table 1, MM1S IGHJ under LymphoTrack is missing a number.

Thank you for this observation. This has been corrected.

Reviewer #2

Multiple myeloma is a hematological disease characterized by unstable genome, various translocations and clonal immunoglobulin variations. This paper describes a novel hybrid-capture approach targeting both ends of V and J segments of immunoglobulins to detect VDJ rearrangements as well as oncogenic translocations. Data are strongly supportive. The manuscript is clear. No additional experiments required in my opinion.

We very much appreciate your review. Thank you for the positive comments on our work.

Reviewer #3

In this paper, Chow et al present their work on the development of a sensitive capture based method to detect V(D)J rearrangements, IGH translocations and mutations in MM samples. The work is mainly technical, and packed with details. This is to be praised, even if some issues arise from disconnects between tables and text in terms of numbers, and should be fixed because they make the results very hard to follow.

Also, while their technical approach is somehow different from others, limitations in cfDNA have been extensively described (Oberle et al 2017, Manzoni et al 2020), and potential advantages of an "all in one" approaches have been published as well by several groups. Therefore, the novelty is relatively low, even if there is some confirmatory value

One important message is that shorter DNA fragments -150bp- may carry limited value to map IGH Tx breakpoints. The authors use cellular DNA to infer implications for cfDNA usage. However, aside from size, also quality and quantity of cfDNA can be a problem, and I would be careful translating findings here to the cfDNA field. It could be quite worse than this.

We agree that there may be significant limitations in translation of this technique to cfDNA as outlined in the discussion. We have elaborated further on these limitations in this revision, excerpted below (**bold** indicates revised text)

These limit of detection findings were also not reproducible with shorter 150bp DNA fragments. A minimal length spanning the junction between structural rearrangements may be required to detect such structural changes. Previous work with T cell receptor V(D)J rearrangements has estimated >99% sensitivity to detect V-J rearrangements with a fragment length of 182bp (Mahé 2016).

*In spite of this, cfDNA and bone marrow findings were highly concordant for V(D)J rearrangements in patients that had clinically detectable disease (Table 2), provided a minimum input of 83ng of cfDNA were input into DNA library preparation. We were not as successful in post-transplant patients with MRD (Suppl. Table 7). In addition to the structural limitations described, this may also be due to significantly decreased ctDNA shedding in cases where myeloma is relatively quiescent. **Therefore detection of V(D)J rearrangements may be limited to use in either genomic DNA or where there is sufficient disease activity to generate the minimum required ctDNA.***

In table S4A, there are some discrepancies in numbers: are there 48 sequences (as in the table) or 47 (as in the text)? Similarly, WGS algorithm called 32 or 30?

There were 48 sequences found as in Supplementary table 4A and WGS algorithm called 32. These have been corrected in the manuscript text.

September 2, 2022

RE: Life Science Alliance Manuscript #LSA-2022-01543-TR

Dr. Trevor J Pugh
Princess Margaret Cancer Centre
MaRS Centre, 101 College Street
PMCRT, Room 9-305
Toronto, ON M5G 1L7
Canada

Dear Dr. Pugh,

Thank you for submitting your revised manuscript entitled "Myeloma Immunoglobulin Rearrangement & Translocation detection through Targeted Capture Sequencing". We would be happy to publish your paper in Life Science Alliance pending final revisions necessary to meet our formatting guidelines.

- please add ORCID ID for secondary corresponding author-they should have received instructions on how to do so
- please add a category for your manuscript to our system
- please consult our manuscript preparation guidelines <https://www.life-science-alliance.org/manuscript-prep> and make sure your manuscript sections are in the correct order
- please add a figure legend for Supplementary Figure 5 to your figure legend section
- please add figure callouts for Figure S6 and Figure S7 to your main manuscript text
- please include the supp. Material file in the supplementary figures
- please include the supp. Methods file into the main manuscript Material and Methods section
- please add a separate approval/ethics statement for the human samples

A. FINAL FILES:

B. MANUSCRIPT ORGANIZATION AND FORMATTING:

Sincerely,

Reviewer #1 (Comments to the Authors (Required)):

The reviewer is satisfied with the revisions. The manuscript is now ready for the next step in publication.

Reviewer #3 (Comments to the Authors (Required)):

the authors have addressed my concerns

September 30, 2022

RE: Life Science Alliance Manuscript #LSA-2022-01543-TRR

Dr. Trevor J Pugh
Princess Margaret Cancer Centre
MaRS Centre, 101 College Street
PMCRT, Room 9-305
Toronto, ON M5G 1L7
Canada

Dear Dr. Pugh,

Thank you for submitting your Resource entitled "Myeloma Immunoglobulin Rearrangement & Translocation detection through Targeted Capture Sequencing". It is a pleasure to let you know that your manuscript is now accepted for publication in Life Science Alliance. Congratulations on this interesting work.

DISTRIBUTION OF MATERIALS:

Again, congratulations on a very nice paper. I hope you found the review process to be constructive and are pleased with how the manuscript was handled editorially. We look forward to future exciting submissions from your lab.

Sincerely,
